# Reinforcement Learning in 20Q Game with Generic Knowledge Bases

**Chun Hei Lo, Luyang Lin**
Department of Systems Engineering and Engineering Management
The Chinese University of Hong Kong
Shatin, Hong Kong
{chlo,lylin}@se.cuhk.edu.hk

## Abstract

20Q, invented by Robin Burgener in 1988, is a computerized game of twenty questions that began as a test in artificial intelligence. The game asks the player to think of something and the system will then try to guess what they are thinking of with twenty yes-or-no questions. Due to the highly flexible and volatile game setting, designing a best questioning strategy for the system is not trivial. In the proposal, we hope to provide possible research directions, particularly under the formulation of the problem as reinforcement learning. We would also investigate on methods and potential challenges of incorporating the use of a knowledge base for the game. The presentation can be found at https://drive.google.com/file/d/1lZ7tTvTFfspJ1P_TEpJvG5TPu0lFSVW6/view?us=sharing

## 1 Problem Definition

20Q consists of two players, a questioner and an answerer (4). The questioner needs to guess the concept thought by the answerer by considering the answerer's replies of the 20 questions asked, where each reply is chosen from one of the three choices of "Yes", "No" or "Unknown". The game starts with the questioner asking a question. Then the two players take turn to answer and ask. The game ends when the 20th question is answered by the answerer.

## 2 Related Work

The questioning strategy is first solved by using a object-question relevance table to rank questions and objects (1). It is further improved by Wu et al. (7) with entropy-based metrics. These method performs tabular computation with rules based on the observed data, which could make them less robust against noisy answers. Zhao and Maxine (8) apply value-based Reinforcement Learning (RL) model to Q20 game, nevertheless, experiments were only conducted on small toy data. Hu et al. (2) proposed a policy-based RL model trained using the data collected from thousands of 20Q games , which shows robustness to noisy answers. Instead of relying on a knowledge base, they have collected a frequency distribution for every person-question pair of 1000 famous people and 500 questions from thousands of real users, which may serve the same purpose as a knowledge base.

## 3 Problem Formulation

The learning of the agent's policy can be formulated into a finite Markov Decision Process, represented by the tuple $\langle S, A, P, R, \gamma \rangle$, where $\gamma \in [0, 1]$ is the reward discount factor for computing long-term return. At time step $t$, the agent is at state $s_t \in S$. It transits to the next state $s_{t+1} \in S$ after taking an action $a_t \in A$ to ask a question according to the policy $\pi_\theta(a|s_t)$ and receiving the answer (Yes/No/Unknown) from the player. After the transition, the agent receives a reward scalar $r_{t+1}$.

# 4 Environment and Data

The environment under which the reinforcement learning algorithm operates is mainly the player, who decides on the concept and answers each of the questions asked by the agent. Rewards are also given to the agent when it succeeds or fails to correctly guess the concept by asking no more than 20 questions.

The game requires data about the concepts to be guessed, otherwise the agent will have to be trained from scratch, where it will be introduced about different concepts paired with the relevant questions and answers. Previous works used either data acquired from thousands of gameplay or a small dataset designed specifically for the game.

We hope to extend the game to a more general setting, where concepts and questions involved are not pre-defined. Following this motivation, we would experiment with different generic knowledge bases, including ConceptNet (3) and Microsoft Concept Graph (6). Both knowledge bases model the hierarchy of and the relationships between millions of concepts collected from difference sources of data.

One of the problems of applying knowledge bases into the game is that the reinforcement learning model may not be learning well if the concept-relation information is sparse, i.e. there are inadequate relations for a concept and/or each relation covers only a few concepts. The detailed implementation of using the knowledge bases and the possible solutions to the knowledge sparsity problem are discussed in the coming section.

# 5 Proposed Approach

The concepts $\mathcal{O}$ to be guessed are one of the concepts in the knowledge bases and the questions to be asked by the questioner $A$ are derived from the relations in the knowledge bases. For example, Microsoft Concept Graph contains the relation triple $\langle \texttt{hamster}, isA, \texttt{small pet} \rangle$, which can be transformed into *hamster* $\in \mathcal{O}$ and *"is it a small pet?"* $\in A$. In general, the game can be conducted on top of any domains over any in the knowledge bases, provided that the concept-relation data is dense enough to distinguish one concept from another using limited number of questions. In this work, we extracted subsets of the knowledge bases' information for our experiments.

We follow Hu et al.'s (2) work for the general implementation of the reinforcement learning framework, with modifications made to extend the game's generalizability. We briefly describe below the different components of the RL framework. Please refer to (2) for the detailed description and implementation.

## 5.1 State Representation

Each state $s_t$ contains the confidence of each concepts at time-step $t$, i.e. $s_t \in \mathbb{R}^{|\mathcal{O}|}$, and $\sum_{i=1}^{n} s_{t,i} = 1$, where $\mathcal{O} = \{o_1, o_2, \ldots, o_n\}$ represents the set of all concepts, and $s_{t,i}$ is the probability that $o_i$ is the user-chosen concept at time-step $t$. There are two choices for the assignment of the initial state $s_0$. It can be the priors of choosing the concept $o_i$ if we assume a prior probability for user choosing the concept, or just a uniform distribution if such assumption is absent.

## 5.2 Transition Dynamics

Given the current state $s_t$ and the answer $x_t$ from the answerer to the question $a_t$, the next state $s_{t+1}$ is computed by $s_{t+1} = s_t \odot \alpha$, where

$$
\alpha = \begin{cases} [R(1, a_t), \ldots, R(|\mathcal{O}|, a_t)] & x_t = \text{Yes} \\ [W(1, a_t), \ldots, W(|\mathcal{O}|, a_t)] & x_t = \text{No} \\ [U(1, a_t), \ldots, U(|\mathcal{O}|, a_t)] & x_t = \text{Unknown} \end{cases}
$$

$R(i, a_t)$, $W(i, a_t)$ and $U(i, a_t) = 1 - R(i, a_t) - W(i, a_t)$ are probabilities of answering "yes", "no" and "unknown" to question $q_{a_t}$ concerning concept $o_i$. Originally they were estimated from the frequencies of the answers from thousands of users to each concept-question pair. Since, our aim is to extend the framework to general question-answering based on a knowledge base, we modify such

probabilities estimation using the information in the knowledge bases, namely:

$$R(i,j) = \begin{cases} r & \langle i, relation_j, concept_j \rangle \in K \\ w & otherwise \end{cases}$$

$$W(i,j) = \begin{cases} w & \langle i, relation_j, concept_j \rangle \in K \\ r & otherwise \end{cases}$$

$$U(i,j) = 1 - R(i,j) - W(i,j)$$

where $K$ is set of relation triples in the knowledge base, $r$ can be considered the reliability of the knowledge base on the relation triple and $w$ can be considered the chance that the relation triple is incorrect as suggested by the knowledge base. This soft assignment of relation triple addresses the problem of incorrect/incomplete information of knowledge bases.

### 5.3 Policy Network

In the policy network, the state $s_t$ is mapped to a probability distribution over all available actions through a neural network with parameter $\theta$: $\pi_\theta(a|s_t) = \mathbb{P}[a|s_t; \theta]$, where $\theta$ is upated to maximize the expected return received. A masked softmax function is applied at the output layer to prevent the model from asking the same question twice.

The long-term return $G_t$ is given by

$$G_t = \sum_{k=0}^{T} \gamma^k r_{t+k+1}$$

where $r_{t+k+1}$ is estimated by $f_r$, a MLP with sigmoid output:

$$r_{t+1} = f_r(s_t, a_t)$$

$f_r$ is trained following loss function:

$$L_1(\sigma) = (R(s_t, a_t; \sigma) - sigmoid(G_t))^2$$

where $\sigma$ is the network parameters.

The policy-based agent is trained using REINFORCE (5) with the loss function:

$$L_2(\theta) = \mathbb{E}_{\pi_\theta}[\log \pi_\theta(a_t|s_t)(G_t - b_t)]$$

where $b_t = \mathcal{V}_\eta(s_t)$ is the estimated expected future reward at state $s_t$. The value network $\mathcal{V}_\eta(s_t)$ is modeled as a MLP and is trained with the objective of minimizing

$$L_3(\eta) = (\mathcal{V}_\eta(s_t) - G_t)^2$$

## 6 Experimental Setup

### 6.1 Model Training with a User Simulator

To allow the model to be trained efficiently, a user simiulator is constructed, which is capable of answering the questions accurately according to the knowledge base. In an episode, the user simulator first selects a concept according to a distribution and it would answer sequentially the 20 questions asked by the RL agent. It finally returns the correctness of the agent's guess and a reward signal is given. Through this interaction between the agent and the user simulator, the agent is trained to select the most proper question to ask each time.

### 6.2 Experimental Results

We apply our approach into two knowledge bases and show our performances in Figure 1. According to ConceptNet, we select two sets. In fig. 1(a), we find that the set with 128 concepts and 64 questions performs much better than the set with 250 concepts and 125 questions. From our observation, the concept-relation information is very sparse in the sampled ConceptNet instances, as the ConceptNet contains several tens of relation between concepts, and some concepts have only a few relations

linked to them. This could pose challenges to the learning of the agent because the user simulator receives inadequate information to many concept-relation instances. We created another data set by sampling concepts and their relations from the Microsoft Concept Graph. Different from ConceptNet, there is only one relation 'is-a', so each concept contains more overlapping relations and each relation are linked to more overlapping concepts. Under this setting with denser data, the win rate improves to more than 0.9 for a test instance. The detailed numbers are shown in fig. 1(b).

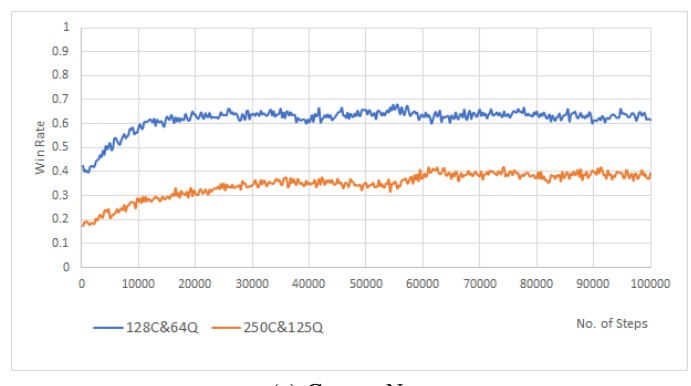

(a) ConceptNet

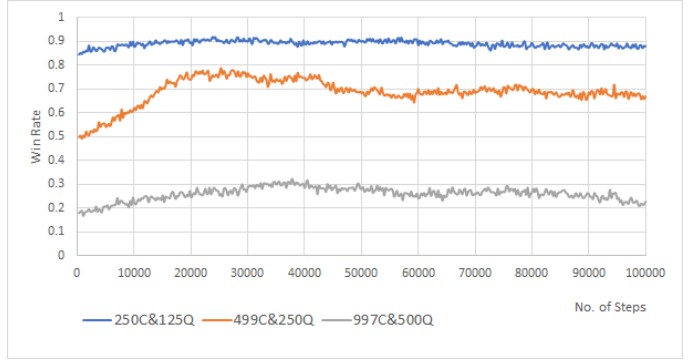

(b) Microsoft Concept Graph

Figure 1: Win Rates with Respect to the Number of Training Steps

| Knowledge Base | Data Set | Win Rate |
|---|---|---|
| ConceptNet | 128C 64Q | 0.68 |
| | 250C 125Q | 0.42 |
| Microsoft Concept Graph | 250C 125Q | **0.92** |
| | 499C 250Q | 0.79 |
| | 997C 500Q | 0.32 |

Table 1: Highest Win Rates on Different Data Sets

Table 2 shows an example of a gameplay where the player chooses the concept *otter*. In this question list, we can find that the questions are not only related to the classification of this animal, but also its special meaning in the real world. For example, otter is also a name of a program. It may help the questioner to guess the concept, and better fit the real game environment.

### 6.3 Conclusion and Future Work

In this work, we have implemented a reinforcement learning framework for the 20Q game under a Markov Decision Process formulation. Much of the implementation follows the work by (2), but we

have extended the application to general question-answering when the relevant knowledge is supplied by a knowledge base. Experiment results show that the questioner agent is indeed able to select the appropriate questions for the final guess of concepts. Nevertheless, it is also important to note that the effectiveness of the training, and thus the performance of the agent depends greatly on the quality of the knowledge base and the denseness of the concept-relation information. Generally, the game is better if more accurate and more dense knowledge is supplied.

There are also opportunities for further experiments on better game modeling. For instance, currently we treat relations that are absent in the knowledge bases as false, i.e. the answers to the corresponding questions are considered "No" with a high probability. One possible solution to that is applying models from the natural language processing community. In particular, automatic hypernym detection would provide the knowledge base with more information regarding the "is-a" relations between concepts. With extra knowledge, a sparse concept-relation matrix should be turned denser and each entry in the table can be estimated more accurately.

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

| No. | Question (*Is your concept a X?*) | Answer |
|-----|-----------------------------------|--------|
| 1   | character                         | Yes    |
| 2   | rare species                      | Yes    |
| 3   | case                              | No     |
| 4   | pattern                           | Yes    |
| 5   | top brand                         | Yes    |
| 6   | symbol                            | No     |
| 7   | exception                         | No     |
| 8   | area                              | No     |
| 9   | carnivore                         | Yes    |
| 10  | common species                    | Yes    |
| 11  | item                              | Yes    |
| 12  | mobile specie                     | Yes    |
| 13  | top predator                      | Yes    |
| 14  | local wildlife                    | Yes    |
| 15  | system                            | No     |
| 16  | form                              | Yes    |
| 17  | option                            | No     |
| 18  | wild creature                     | No     |
| 19  | exotic species                    | No     |
| 20  | program                           | Yes    |

Table 2: The 20 questions and answers to guessing the concept *otter*

