# OpenReview forum: "Reinforcement Learning in 20Q Game with Generic Knowledge Bases"
_CUHK.edu.hk/2021/Course/IERG5350_

### Official Review · AnonReviewer2 · 2020-12-15
**review feedback**

**Rating:** 8
**Confidence:** 3

**Review:**

Quanlity -
The paper is well structured and overall quality is decent.

Clarity -
The overall clarity is excellent. There is clear problem definition, how the problem is formulated in RL problem. Results comparision and future improvement proposal are provided.

Originality -
Since it's applying existing knowledge graph with modification to generalize, I'd think it's decent on this.

Significance -
Maybe I don't fully understand why this is a problem worth solving. I'd think it's acceptable level on this.

---

### Official Review · AnonReviewer1 · 2020-12-20
**Well-defined problem solved with the help of  RL**

**Rating:** 7
**Confidence:** 4

**Review:**

General:
The paper tried to explore the application of reinforcement learning on an interesting game named 20Q. To maximize the reward, they used the policy-based method to train their agent and proved it works to help their agent to get better performances on the data sampled from the ConceptNet and Microsoft Concept Graph.


Evaluation of the quality: This paper proposed to use a policy-based method for the 20Q game. After the training, their agent can have a 90% win rate in data sampled from Microsoft Concept Graph.

Clarity:
The paper is clearly written and well organized. The state and transition dynamics of their project are well-defined. But there is not enough analysis of their results.

Originality:
They collected their own data set from the ConceptNet and Microsoft Concept Graph. In addition, they defined their objective function clearly.

Significance:
The paper tried a policy-based reinforcement learning algorithm on the 20Q game. But they need to show the reason why they used this method. More experiments are needed in the same data set. For example, experiments to show the comparison between their method and the value-based method they mentioned in the related work.

Pros:
1.	This paper is well-organized.
2.	This paper has a clear problem formulation.
3.	The table and graphs show the performance of the proposed approach is good.

Cons:
1.	There are a few grammar mistakes, for instance, “These method performs tabular computation with rules based on the observed data”, the “method” should be “methods”.
2.	They need more experiments.

---

### Official Review · AnonReviewer3 · 2020-12-20
**Good submission**

**Rating:** 6
**Confidence:** 3

**Review:**

This paper focuses on the classical 20Q game. The authors use reinforcement learning to solve this problem. Experiments have been conducted to show the effectiveness of the proposed method.

The writing of this paper is clear and easy to follow. The equation formulation and notations are also clear. Experiments also show the effectiveness of the proposed problem.

However, the importance of this problem has not been clearly stated in the paper. It is suggested that the authors could clarify this point in the next version.